# Clinical and Immunologic Efficacy of the Recombinant Adenovirus Type-5-Vectored (CanSino Bio) Vaccine in University Professors during the COVID-19 Delta Wave

**DOI:** 10.3390/vaccines10050656

**Published:** 2022-04-21

**Authors:** Santos Guzmán-López, Armine Darwich-Salazar, Paola Bocanegra-Ibarias, Daniel Salas-Treviño, Samantha Flores-Treviño, Eduardo Pérez-Alba, Laura M. Nuzzolo-Shihadeh, Edelmiro Pérez-Rodríguez, Adrián Camacho-Ortiz

**Affiliations:** 1Rector of the Universidad Autónoma de Nuevo León, Torre de Rectoría en Ciudad Universitaria, San Nicolás de los Garza 66451, Mexico; santos.guzmanl@uanl.mx; 2Department of Infectious Diseases, School of Medicine, University Hospital Dr. José Eleuterio González, Universidad Autónoma de Nuevo León, Monterrey 64460, Mexico; armine@darwich.mx (A.D.-S.); pbocanegrai@uanl.edu.mx (P.B.-I.); daniel.salastr@uanl.edu.mx (D.S.-T.); samantha.florestr@uanl.edu.mx (S.F.-T.); eduardo.perezlb@uanl.edu.mx (E.P.-A.); laura.nuzzolos@uanl.edu.mx (L.M.N.-S.); 3School of Medicine, Hospital Universitario Dr. José Eleuterio González, Universidad Autónoma de Nuevo León, Monterrey 64460, Mexico; edelmiro.perezrd@uanl.edu.mx

**Keywords:** COVID 19, vaccine, SARS-CoV-2, immunity, adenovirus vector vaccine

## Abstract

Information regarding the efficacy of the recombinant adenovirus type-5-vectored (CanSino Bio) vaccine against the COVID-19 disease in a real-life setting is limited. A retrospective cohort study was carried out in the teaching university community of the metropolitan area of Monterrey, Mexico, through a four-section survey, and during the COVID-19 delta wave. Determination of IgG antibodies against SARS-CoV-2 spike (S) protein was performed in a subset of participants vaccinated with CanSino Bio. A total of 7468 teachers responded to the survey, and 6695 of them were fully vaccinated. Of those, 72.7% had CanSino Bio, 10.3% Pfizer, 8.4% AstraZeneca, 1.2% Moderna, and 2.7% others. Symptomatic breakthrough infections were recorded in those vaccinated with CanSino Bio (4.1%), AstraZeneca (2.1%), and Pfizer (2.2%). No difference was found between CanSino Bio and other vaccines regarding hospitalization, the need for mechanical ventilation, and death. For CanSino Bio recipients, anti-S antibodies were >50 AU/mL in 73.2%. In conclusion, primary breakthrough symptomatic infections were higher in the CanSino vaccinated group compared to other brands. Individuals with a previous infection had higher antibody levels than those who were reinfected and without infection. A boosted dose of CanSino is recommended for those individuals without a previous infection.

## 1. Introduction

From the beginning of the SARS-CoV-2 pandemic, multiple pharmaceutical companies undertook the task of developing vaccines to aid against the COVID-19 disease. To date, 48.7% of the world’s population has received at least one dose of a COVID-19 vaccine; 6.87 billion doses have been administered globally and 25.74 million are now administered each day [1]. These vaccines have been developed using novel methods, such as messenger RNA technology, which increases the volume and speed of production compared to other types of vaccines, while improving product stability and generating strong immune responses [2]. Other vaccines use existing methods facilitating their large-scale production, such as inactivated virus-based vaccines that use previously attenuated viruses in such a way that they do not cause disease, by means of harmless protein fragments or protein structures that mimic the virus or viral vector vaccines, using an innocuous genetically modified virus [2].

The World Health Organization (WHO, Geneva, Switzerland) has communicated the effectiveness and safety of vaccines through statements [3], for example, the effectiveness of the Pfizer/BioNTech (95%), Moderna (94%), Sputnik V (92%), Novavax (89.3%), Sinopharm (79.34%), and AstraZeneca (70%) vaccines, among others. To date, each country’s government has been responsible for their approval, entry, and administrative policies [3].

In the case of Latin America, governments are responsible for purchasing vaccines, depending on their financial capacity and as a part of public health policy. The adequate distribution was proposed according to the Pan American Health Organization (PAHO, Washington, DC, USA) [4] through strategic and informed planning, and its application was carried out in different phases, considering healthcare personnel as priorities followed by other healthcare providers. Teachers were the second large-scale group that was vaccinated in some countries [5]. In Mexico, for example, vaccination with CanSino (CanSino Biologics Inc., Tianjin, China) started on April 2021 for school teachers of all grades, from preschool to college [6]. Concerns remain about the efficacy of some of those vaccines including CanSino because of the absence of published date on phase III studies. For educational personnel, the CanSino vaccine was selected since it offered the advantage that it only required a single dose, allowing teachers to be able to return to the classrooms. This decision was part of the strategy outlined from the National Vaccination Plan that had the support of the Technical Advisory Group of Vaccination [6].

Owing to limited published information regarding the effectiveness of the CanSino vaccine in a real-life setting, we analyzed clinical and immunological data on a cohort of vaccinated faculty staff and compared them with other vaccines.

## 2. Materials and Methods

### 2.1. Ethics

The trial protocol was reviewed and approved by the local Research and Ethics Committee of the University Hospital “Dr. José Eleuterio González” (approval number IF21-00016). This work has been carried out in accordance with the Code of Ethics of the World Medical Association (Declaration of Helsinki: ethical principles for medical research involving human subjects).

### 2.2. Study Design and Participants

This was a retrospective cohort study to which the entire teaching community of the Autonomous University of Nuevo León (UANL, San Nicolás de los Garza, Nuevo León, México), located in the metropolitan area of Monterrey, México, was invited to participate. If compliant, a brief 4-section survey that included demographic data, history of vaccination against SARS-CoV-2, history of infection by the latter, and a consent for blood sampling was collected through the SIASE system (Integrated System for the Administration of Educational Systems, Universidad Autónoma de Nuevo León, Monterrey, Nuevo León, México), an electronic platform to which all registered teachers of the UANL have regular access. The survey was open from 16 to 25 August 2021. The study participants were professors at high schools, technical preparatory schools, university, and postgraduate courses belonging to the UANL. Eligible participants were adults aged 18 years or older. To be included, participants needed to be able to sign an informed consent and be able and willing to complete all the scheduled study processes. Potential participants that had heterologous vaccination by the time of the study were excluded. Participants were classified by self-reported history of COVID-19 before and/or after vaccination and by the brand of vaccine received.

### 2.3. Blood Sampling and Antibody Measurement

All the participants that agreed to blood sampling were invited to do so via e-mail, but only a subset of them accepted to do so. During 13–17 September 2021 and 20–24 September 2021, a 5 mL blood sample from each participant was taken and subjected to a chemiluminescent microparticle immunoassay used for the qualitative and quantitative determination of IgG antibodies against SARS-CoV-2 spike (S) protein (Architect i1000SR, Abbott Laboratories, Chicago, IL, USA). On the day of blood sampling, patients were reinterrogated for new heterologous or homologous vaccine doses and for COVID-19 history.

### 2.4. Definitions

Complete vaccination was defined as a participant who at the time of the survey had ≥30 days after receiving a single-dose vaccine or those who received a two-dose vaccine scheme and ≥7 days had passed after the second dose. Incomplete vaccination was considered as any of the participants who had received at least one dose and did not comply with the time frames described above. Participants were classified by the brand of vaccine they had received: CanSino (CanSino Biologics Inc., Tianjin, China), Pfizer/BioNTech (Mainz, Germany), AstraZeneca/University of Oxford, Sputnik V (Gamaleya Research Institute, Moscow, Russia), Janssen/J&J (Janssen Pharmaceuticals/Johnson & Johnson, Titusville, NJ, USA), CoronaVac (Sinovac Life Sciences, Beijing, China), and Moderna (TX, Inc., Cambridge, MA, USA).

The participants were also divided into four groups according to self-reported COVID-19 infection: (1) breakthrough infection, those who were infected for the first time after being vaccinated, (2) reinfection, those who were infected before and after receiving their vaccine, (3) previous infection, those who were only infected before the vaccination, and (4) without infection, those who were not infected before or after the vaccine.

### 2.5. Statistical Analysis

The data were coded and cleaned to ensure their integrity and consistency. Central tendency measures were used for grouping the data. Chi-square and Fisher’s exact tests were used for dichotomous variables, and the Student’s t-test was used for the comparison of means of continuous variables.

The sample size for serologic testing was determined based on the total population of registered teachers that completed the survey with a confidence level of 95% and an error margin of 5. We calculated a number of 360 random subjects for serological sampling. A *p*-value ≤ 0.05 was accepted as a statistically significant difference and SPSS version 20 was used in addition to MedCalc^®^ for statistical analysis.

## 3. Results

### 3.1. Clinical and Epidemiologic Data

The total population of teachers registered in the SIASE platform was 8392 (100%), of which 7468 (88.9%) responded to the survey. The mean age of the participants was 43 years (range 18–87), with 4153 (55.6%) individuals aged 18–44 years, 1763 (23.6%) aged 45–54 years, and 1552 (20.7%) aged 55 years or older. Self-reported comorbidities were 1143 (15.3%) obesity, 800 (10.7%) hypertension, 493 (6.6%) diabetes, 192 (2.5%) asthma, 121 (1.6%) autoimmune disease, 46 (0.61%) ischemic heart disease, 45 (0.60%) cancer, and 11 (0.14%) chronic obstructive pulmonary disease.

At the time of the survey, those who were vaccinated against SARS-CoV-2 were 7150 (85.2%), of whom 6695 (79.7%) were classified as having complete vaccination and 455 (5.4%) as incomplete vaccination. We excluded 117 participants from the complete vaccination group and 112 from the incomplete vaccination group due to lack of information (See Figure 1), leaving only 6921 participants for clinical analysis. Outcomes for the incomplete vaccination group and unvaccinated participants are available in Appendix A, respectively. Of those, 2562 participants consented to be invited for serologic sampling. At the time of the survey, the average (and SD) number of days since vaccination was as follows: CanSino 115 (±28.2), Pfizer/BioNTech 106 (±49.5), AstraZeneca/Oxford 85 (±42.4), Sputnik V 85 (±24.7), Janssen/J&J 108 (±45.2), CoronaVac 67 (±43.7), and Moderna 113 (±36.4).

The percentage of primary breakthrough symptomatic infection was higher in the CanSino vaccinated group compared to the other brands (4.1% vs. 1.6%, *p* = 0.0001). Requirement of hospitalization, supplemental oxygen, mechanical ventilation, and death were not statistically different among these groups (see Table 1 and Appendix A). The incidence of symptomatic reinfection was not different between the CanSino group compared to other brands, and neither were the requirement for hospitalization, supplemental oxygen, mechanical ventilation, and death.

### 3.2. Serologic Results

Of the 271 completely vaccinated participants with CanSino, 67 had a breakthrough infection, and also had a mean (and SD) S antibody value of 16,250.5 AU/mL (±28,601.4); three individuals had reinfections and an S antibody value of 3373.1 AU/mL (±2250.3); 56 individuals from the previous infected group had a mean S antibody value of 4628.9 AU/mL (±5900.7); 145 of the participants without previous infection had a mean S antibody value of 1237.2 AU/mL (±3457.3). For details regarding the administered vaccine and self-reported COVID-19 group, please refer to Appendix A. Using the manufacturer’s cut-off point, of the total sampled from the CanSino group, a total of 46 (16.9%) participants were considered negative (less than 50.0 AU/mL), of which 5 self-reported being hypertensive, 5 suffered from obesity, 3 had diabetes, 2 asthma, and 1 autoimmune disease.

## 4. Discussion

The administration of COVID-19 vaccines in Mexico was accomplished using several types of available vaccines in the country, which are currently government-approved [7]. BNT162b2 (Pfizer/BioNTech), an mRNA vaccine, was first administered in the country in late December to prioritized population groups, starting with COVID-19-related health personnel [8]. Four non-replicating recombinant adenovirus vectors including Ad5-nCoV (CanSino Biologics Inc), AZD1222 (AstraZeneca/University of Oxford), Ad26.COV2.S (Janssen Pharmaceuticals/Johnson & Johnson), and Sputnik V (Gamaleya Research Institute) [9,10,11,12], the inactivated whole-virus CoronaVac (Sinovac Life Sciences) vaccine [13], and mRNA-1273 (Moderna TX, Inc) vaccine [10] have also been used to vaccinate the Mexican population.

The CanSino vaccine was offered exclusively to educational personnel in April 2021 [6,8], with the hopeful aim of opening schools soon after. However, the effectiveness of the CanSino vaccine against SARS-CoV-2 infection or COVID-19 severity in different populations worldwide is not completely known. The efficacy of the CanSino vaccine in its early stages showed increased SARS-CoV-2 specific antibodies 28 days post-vaccination in healthy adults [9], and significant immune-induced responses after a single dose in both adults aged 18 years or older [14] and children and adolescents aged 6–17 years [15]. However, the lack of phase III published studies is still a cause of concern. Therefore, we analyzed clinical and immunological data on a cohort of teaching participants vaccinated with CanSino and other vaccines.

The total population of registered teachers at our institution currently are 8392, in which the most common self-reported comorbidities were obesity (15.3%), hypertension (10.7%), and diabetes (6.6%). Up to 7150 (85.2%) participants received at least one dose of a COVID-19 vaccine, 6695 (79.7%) had complete vaccination, and 6921 participants were eligible for clinical analysis. The percentage of primary breakthrough symptomatic infections (i.e., vaccinated individuals who can be infected and develop COVID-19 symptoms) was higher in the CanSino vaccinated group compared to the other brands (4.1% vs. 1.6%, *p* = 0.0001). However, the incidence of symptomatic reinfection, the requirement of hospitalization, supplemental oxygen, mechanical ventilation, and death were not different between the CanSino group compared to other brands.

The measurement of IgG antibodies against the SARS-CoV-2 spike region that binds to the receptor (RBD, receptor binding domain) is used as an aid in the diagnosis of SARS-CoV-2 infection [16], as well as to evaluate the immune status of infected people and monitor the antibody response in people vaccinated against the COVID-19 virus, through quantitative determination, as higher antibody titers correlate with higher vaccine efficacy [17]. Participants with breakthrough infections had higher anti-S antibodies than all three groups. Interestingly, individuals with a previous infection had higher antibody levels than those who were reinfected and those without infection (mean ± SD: 4628.9 ± 5900.7 vs. 3373.1 ± 2250.3 vs. 1237.2 ± 3457.3 AU/mL).

Only two studies have evaluated the efficacy of the CanSino vaccine in Mexican populations [18,19]. Our results are similar to another study, in which higher neutralizing antibodies were detected in CanSino-vaccinated individuals with previous infection compared to those without previous infection (median 98% vs. 72%, *p* < 0.0001) [19]. In addition, the neutralization percentage was higher in individuals with previous COVID-19 before vaccination than those without prior COVID-19 after vaccination (*p* < 0.001) [19], which would have been interesting to evaluate in our population.

The assessment of anti-S1 IgG antibody generation in 61 people vaccinated with Pfizer/BioNTech and 54 people vaccinated with CanSino showed a higher seronegative percentage in participants vaccinated with CanSino (11.11% vs. 1.64% after Pfizer/BioNTech two-dose administration) [18]. In another study, 17 (7.4%) individuals without previous infection did not have the presence of neutralizing antibodies after the CanSino vaccination [19]. Similarly, in our study population, 46 (16.9%) participants vaccinated with CanSino showed negative anti-S antibody titers.

In a previous study, the CanSino vaccine generated a lower number of S1 IgG antibodies compared to Pfizer/BioNTech (mean 4.19 vs. 8.01). People with previous COVID-19 infection had higher S1 IgG antibody levels with both vaccines (mean index of 11.49 in Pfizer/BioNTech and 7 in CanSino) compared to participants without previous infection (mean of 7.89 in Pfizer/BioNTech and 3.9 in CanSino) [18]. We were not able to compare the antibody levels in our population with other types of vaccines as our faculty was almost exclusively vaccinated with CanSino. However, the presence of anti-S antibodies was detected up to 115 days after CanSino vaccination.

The highest COVID-19 peak in Mexico occurred on 12 August 2021, with 24,975 new daily cases and 608 deaths [20], during the Delta variant wave. Active national genomic surveillance showed that the SARS-CoV-2 Delta variant strongly predominated in the northeastern part of Mexico during the months of June through October 2021. The B.1617.2 variant rapidly rose from 7.8% by mid-June 2021 [21] to 78.1% by July 28 [22] and 97% by August 25 (of which 46.2% belonged to B.1.617.2 and 41.7% to AY.12) [23], maintaining high frequencies with 100% of all sequencing corresponding to the Delta variants [24].

We acknowledge some limitations to the study. First, the fact that described events were self-reported and not confirmed by PCR testing. This may be one of the causes why breakthrough infection reports are higher in the reported literature than they were in our study (5% for Pfizer/BioNTech and 26% for AstraZeneca/Oxford) [25,26]. As the survey was specific for confirmed diagnosis only, participants were only reinterrogated when they came in for blood work and not subjected to PCR testing. Antibodies against the S protein do not necessarily correlate with an adequate immunologic response, as a negative result does not necessarily indicate non-protection. However, this test was readily available in our facility so we used this marker as a partial reflection of the immunological response.

## 5. Conclusions

The incidence of symptomatic reinfection, the requirement of hospitalization, supplemental oxygen, mechanical ventilation, and death were not different between individuals vaccinated with CanSino compared to Pfizer/BioNTech, AstraZeneca/Oxford, Sputnik V, Janssen/J&J, CoronaVac, and Moderna. However, the percentage of primary breakthrough symptomatic infections was higher in the CanSino vaccinated group compared to other brands. Participants with breakthrough infections had higher anti-S antibodies than all three groups. Individuals with a previous infection had higher antibody levels than those who were reinfected and without infection. In addition, 16.9% of individuals vaccinated with CanSino were seronegative. Therefore, a boosted dose of CanSino is recommended for those individuals without a previous infection.

## Figures and Tables

**Figure 1 vaccines-10-00656-f001:**
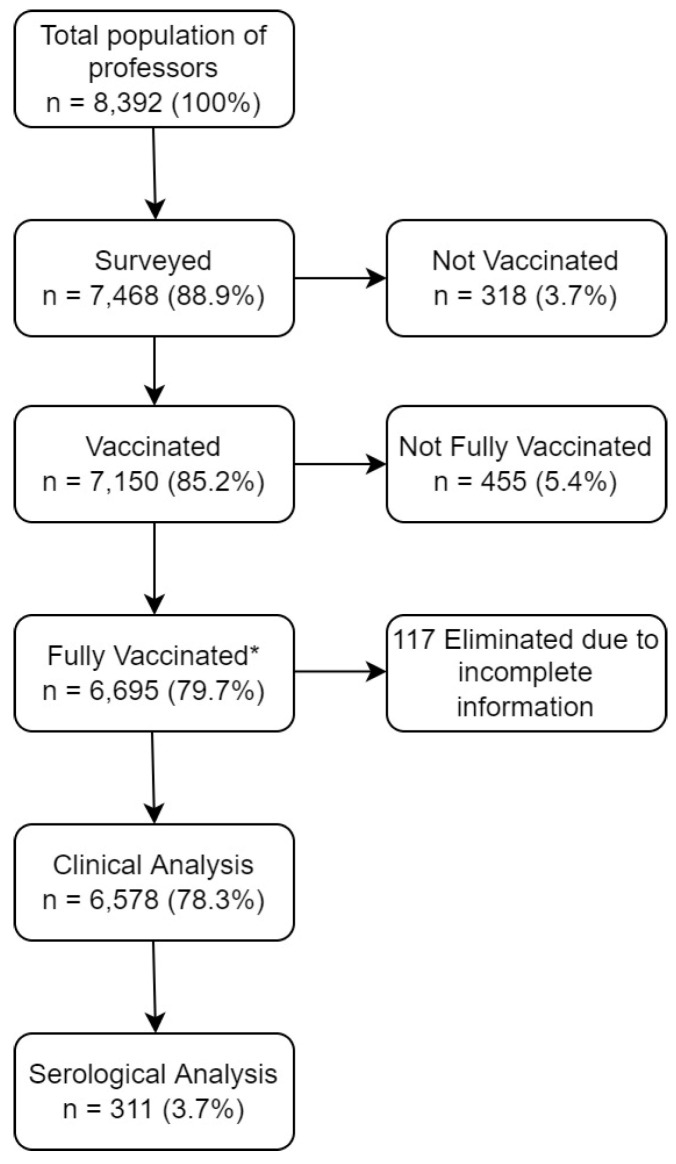
Diagram of the studied population. * Fully vaccinated: those who received a single dose vaccine ≥30 days, or those who received a 2 dose-scheduled vaccine and ≥7 days after the last dose was administered.

**Table 1 vaccines-10-00656-t001:** Clinical characteristics of patients presenting COVID-19 breakthrough infections and reinfections after full vaccination.

After Vaccination	CanSino *n* = 5360 *n* (%)	Pfizer/ BioNTech *n* = 619 *n* (%)	AstraZeneca/ Oxford *n* = 466 *n* (%)	Sputnik V *n* = 5 *n* (%)	Janssen/J&J *n* = 58 *n* (%)	CoronoVac *n* = 97 *n* (%)	Moderna *n* = 90 *n* (%)	*p*-Value *
Primary breakthrough symptomatic infection								
Total	224 (4.17)	14 (2.26)	10 (2.14)	0 (0.00)	2 (3.44)	1 (1.03)	2 (2.22)	<0.001
Outpatient	216 (4.02)	14 (2.26)	10 (2.14)	0 (0.00)	2 (3.44)	1 (1.03)	2 (2.22)	<0.001
Hospitalization	8 (0.14)	0 (0.00)	0 (0.00)	0 (0.00)	0 (0.00)	0 (0.00)	0 (0.00)	0.21
Supplemental oxygen	7 (0.13)	0 (0.00)	0 (0.00)	0 (0.00)	0 (0.00)	0 (0.00)	0 (0.00)	0.20
Mechanical ventilation	1 (0.02)	0 (0.00)	0 (0.00)	0 (0.00)	0 (0.00)	0 (0.00)	0 (0.00)	0.57
Death	1 (0.02)	0 (0.00)	0 (0.00)	0 (0.00)	0 (0.00)	0 (0.00)	0 (0.00)	0.57
Breakthrough symptomatic reinfection from previously infected participants								
Total	15 (0.27)	1 (0.16)	2 (0.42)	0 (0.00)	0 (0.00)	0 (0.00)	0 (0.00)	0.64
Outpatient	14 (0.26)	1 (0.16)	2 (0.42)	0 (0.00)	0 (0.00)	0 (0.00)	0 (0.00)	0.07
Hospitalization	1 (0.01)	0 (0.00)	0 (0.00)	0 (0.00)	0 (0.00)	0 (0.00)	0 (0.00)	0.57
Supplemental oxygen	1 (0.01)	0 (0.00)	0 (0.00)	0 (0.00)	0 (0.00)	0 (0.00)	0 (0.00)	0.57
Mechanical ventilation	0 (0.00)	0 (0.00)	0 (0.00)	0 (0.00)	0 (0.00)	0 (0.00)	0 (0.00)	NA
Death	0 (0.00)	0 (0.00)	0 (0.00)	0 (0.00)	0 (0.00)	0 (0.00)	0 (0.00)	NA

* Significant *p*-values are remarked in bold letters. NA: not applicable.

## Data Availability

Not applicable.

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
