# Peer review of "Clinical and Immunologic Efficacy of the Recombinant Adenovirus Type-5-Vectored (CanSino Bio) Vaccine in University Professors during the COVID-19 Delta Wave"

_vaccines, 2022, doi:10.3390/vaccines10050656_

Round 1

Reviewer 1 Report

Guzman-Loprez et al report on the clinical and immunological efficacy of CanSino Bio vaccine. They present a representative study group of 7468 teachers. The anamnestic data included age, gender and self-reported comorbidities. The authors report an increased incidence of symptomatic breakthrough infections after CanSino vaccination compared to other brands. In conclusion, they recommend booster vaccination with CanSino vaccine. These results are important and merit publication.

The study group is well characterized, however the presentation of data is overall disappointing. In fact, from the available data the authors show only the breakthrough data in sufficient detail. Unfortunately, the conclusions from this table appear over-interpreted since they find hospitalization cases exclusively after CanSino vaccination due to the overall low case number. Consequently, I would propose a more detailed version of the table including age and comorbidity. Additionally, the authors should distinguish between fully vaccinated and incompletely vaccinated participants.

The immunological part is in total somewhat nebulous. 2562 participants consented serologic sampling (p3, lane 135), 311 were investigated (Fig. 1) and 271 were reported (p5, lane 155)?? On page 3, lane 107, the authors define four groups which should appear in the report of the serological analysis in order to draw any conclusions from these results. Overall, the serological data should appear as a comprehensive table considering this definition.

Minor comments:

The percentage of breakthrough infections is not consistent between abstract, text and table. Please check.

The authors should add a definition of “symptomatic breakthrough infection” to the methods part. This must not necessarily be a PCR confirmation by the own lab but can also be a self-reported diagnosis of SARS-CoV-2 infection.

The discussion should include a remark how far the observed incidence of breakthrough infection fits other studies with AstraZeneca or Pfizer vaccine. Is the breakthrough incidence in the same range?

Author Response

Dear Reviewer

We deeply appreciate the revisions and suggestions made. We have made the following changes.

Suggestion 1: I would propose a more detailed version of the table including age and comorbidity. Additionally, the authors should distinguish between fully vaccinated and incompletely vaccinated participants.

Response 1: Table 1 has been changed to present only completely vaccinated participants and a new Supplementary Table 1 is included regarding data from partially vaccinated participants. Supplementary Table 3 addresses participant's data regarding age, gender, and comorbidities.

Suggestion 2: the authors define four groups which should appear in the report of the serological analysis in order to draw any conclusions from these results. Overall, the serological data should appear as a comprehensive table considering this definition.

Response 2: We have clarified that all the participants that agreed to blood sampling were invited to do so via e-mail, but only a subset of them accepted to do so. The information gathered from those participants organized by vaccine type and infection group are included in Supplementary table 4.

Suggestion 3: The percentage of breakthrough infections is not consistent between abstract, text and table. Please check.

Response 3: Data has been corrected. Thank you.

Suggestion 4: The authors should add a definition of “symptomatic breakthrough infection” to the methods part.

Response 4: Details were added in page 3 line 108-109.

Suggestion 5: The discussion should include a remark how far the observed incidence of breakthrough infection fits other studies with AstraZeneca or Pfizer vaccine. 

Response 5: On page 6 line 239-241 we added information regarding the solicited remark.

Reviewer 2 Report

Dear Authors,

The manuscript titled "Clinical and Immunologic Efficacy of the Recombinant Adenovirus Type-5-vectored (CanSino Bio) Vaccine in University Professors during the COVID-19 Delta Wave" is interesting but some lacks are present and should not be considered for publication in the present form.

Here my major concern:

1-In figure 1, the 117 eliminated people from your analysis, seem to be excluded from the "clinical analysis" group.

2- Since the authors performed sierological analysis, it should be fine to add the obtained results also in a table, even if there is no comparison with other vaccination groups. Furthermore, it should also be fine to show the results by gender, age, pathologies and so on.

3- Did the Authors have any followup concerning the sierological analysis and/or the clinical characteristics after 3-6 month? It should be interesting to know how the immunological response after a long period after vaccination.

4-Finally, a comparison with not vaccinated (with and without covid infection), not fully vaccinated and fully vaccinated participants concerning the clinical and immunological response should be performed in order to better underline the efficacy of CanSino vaccine.

Author Response

Dear Reviewer

We deeply appreciate the revisions and suggestions made. We have made the following changes.

Suggestion 1: 1-In figure 1, the 117 eliminated people from your analysis, seem to be excluded from the "clinical analysis" group.

Response 1: Thank you for the observation. Figure 1 has been edited for clarity.

Suggestion 2: Since the authors performed sierological analysis, it should be fine to add the obtained results also in a table, even if there is no comparison with other vaccination groups. Furthermore, it should also be fine to show the results by gender, age, pathologies and so on.

Response 2: The authors have added Supplementary tables 3 and 4.

Suggestion 3: Did the Authors have any followup concerning the sierological analysis and/or the clinical characteristics after 3-6 month? It should be interesting to know how the immunological response after a long period after vaccination.

Response 3: We agree that it would have been an interesting point, but the authorization submitted to the ethics committee didn't include an additional serological follow-up.

Suggestion 4: A comparison with not vaccinated (with and without covid infection), not fully vaccinated and fully vaccinated participants concerning the clinical and immunological response should be performed in order to better underline the efficacy of CanSino vaccine.

Response 4: Supplementary tables 1 and 2 have been added to address the clinical outcomes of unvaccinated and partially vaccinated participants.

Round 2

Reviewer 1 Report

The authors have adressed all comments and added a more detailed data analysis within the supplementary files.